# Mini-Intein Structures from Extremophiles Suggest a Strategy for Finding Novel Robust Inteins

**DOI:** 10.3390/microorganisms9061226

**Published:** 2021-06-05

**Authors:** Mimmu K. Hiltunen, Hannes M. Beyer, Hideo Iwaï

**Affiliations:** Institute of Biotechnology, HiLiFE, University of Helsinki, P.O. Box 56, 00014 Helsinki, Finland; mimmu.hiltunen@helsinki.fi (M.K.H.); hannes.beyer@uni-duesseldorf.de (H.M.B.)

**Keywords:** protein splicing, intein, crystal structure, hyperthermophile, protein engineering

## Abstract

Inteins are prevalent among extremophiles. Mini-inteins with robust splicing properties are of particular interest for biotechnological applications due to their small size. However, biochemical and structural characterization has still been limited to a small number of inteins, and only a few serve as widely used tools in protein engineering. We determined the crystal structure of a naturally occurring Pol-II mini-intein from *Pyrococcus horikoshii* and compared all three mini-inteins found in the genome of *P. horikoshii*. Despite their similar sizes, the comparison revealed distinct differences in the insertions and deletions, implying specific evolutionary pathways from distinct ancestral origins. Our studies suggest that sporadically distributed mini-inteins might be more promising for further protein engineering applications than highly conserved mini-inteins. Structural investigations of additional inteins could guide the shortest path to finding novel robust mini-inteins suitable for various protein engineering purposes.

## 1. Introduction

Self-splicing protein introns (inteins) are genetic elements that are translated with their host proteins [1,2]. After translation, the inteins catalyze their own excision and re-ligation of flanking protein regions (called exteins) with a peptide bond, resulting in active mature host proteins [1,3]. Inteins are often considered selfish genetic elements, found within coding DNA regions, because their removal generally does not affect the fitness of the host organisms [3]. However, specific regulatory functions of inteins have been proposed for some inteins [4,5]. 

Inteins are found in all three domains of life and are often considered to have ancient origins that predate the separation of prokaryotes and eukaryotes [4,5,6,7,8]. A recent analysis of an annotated genome database revealed that up to 16 protein hosts with inteins were found for nearly half of all archaea analyzed, whereas only one percent of eukaryotes contain inteins [6,7]. Inteins are also found among various extremophiles, such as thermophiles, halophiles, and acidophiles. Inteins reside in various host proteins with diverse functions [6,7]. However, there is a bias towards proteins involved in DNA metabolisms, such as DNA polymerases, topoisomerases, and ribonucleotide reductases [6,7]. The biased distribution of inteins to specific proteins, and their insertions at active sites, has led to the hypothesis that some inteins have adapted to become environmental sensors that play regulatory roles by protein splicing conditionally under certain conditions, such as high temperature, salinity, and redox states [4,5,6,7]. 

The closely related thermophiles *P. abyssi* (*Pab*) and *P. horikoshii* (*Pho*) both contain 14 inteins in their genome, while the halophile *Haloquadratum walsbyi* contains 15 inteins (Appendix A) [6]. These three extremophiles are intein-rich because up to 19 inteins have been identified (Appendix A) [6,7]. However, the distribution and size of these inteins are not highly conserved, even among thermophilic archaea of the same genus. Presumably, horizontal gene transfers (HGT) that occurred during evolution contributed to this variation, counterbalanced by the degeneration events of the nested homing endonuclease domains (HENs) [3]. Due to these evolutionary events, the structure and protein-splicing activities of inteins might represent the evolutionary history of each intein [9]. 

Despite the obscure biological/physiological roles of inteins, their enzymatic mechanisms catalyzing protein-splicing reactions have opened a new horizon in protein chemistry. Utilizing the protein splicing activities of inteins bears a repertoire of potential applications in several areas, including *in vivo* protein engineering, protein purification, and modification. Indeed, intein-mediated chemical reactions have increasingly been incorporated as practical tools in the fields of protein engineering, synthetic biology, and biotechnology [10,11]. For example, inteins from extremely halophilic archaea have been demonstrated to control protein-splicing reactions with salt concentrations, which has enabled the engineering of a salt-inducible self-cleaving tag for protein purification [12]. Thus, inteins from extremophiles might have great potential for the development of unique biotechnological tools. 

Inteins often contain a nested active or inactive HEN, which presumably plays an essential role in the HGT of intein genes [3,6,13]. There is a class of inteins lacking the HEN domain, so-called mini-inteins. Due to their reduced complexity, naturally-occurring mini-inteins lacking HEN domains have been of particular interest for protein engineering to develop biotechnological applications [14,15]. Although the protein-splicing HINT (Hedgehog/INTein) and DNA-processing HEN domains likely function independently of each other [1], attempts to engineer mini-inteins by removing the HEN have revealed a more complicated relationship between the two domains [9,15,16,17]. Some engineered mini-inteins retain their splicing activity, while others lose it completely [9,14,15,16]. 

Currently, inteins have not been systematically selected and tested biochemically for the robustness of their protein splicing activity, because their splicing activities are not predictable without experimental assessments. Thus, strategies for choosing promising inteins from sequence databases would be highly desirable in order to best advance future protein engineering. 

As the first step towards a rational approach to identify robust inteins for protein engineering, we turned our attention to the 14 inteins identified in the genome of *P. horikoshii.* Three of these inteins can be classified as mini-inteins (≤200 residues). These are the *Pho*RadA, *Pho*CDC21-1, and *Pho*Pol-II inteins, consisting of 172, 170, and 166 residues, respectively. The structural and biochemical characterizations of the *Pho*RadA and *Pho*CDC21-1 inteins have been previously reported [17,18]. In this work, we determined the 1.48-Å resolution structure of the *Pho*Pol-II intein, the smallest intein found in *P. horikoshii*, and compared the selected inteins of the *Pyrococcus* genus with two other mini-inteins in *P. horikoshii.* The structural comparison and accumulated biochemical data might serve as a practical compass in the quest for robust mini-inteins from genomic sequence data. 

## 2. Materials and Methods

### 2.1. Cloning and Production of PhoPol-II Intein

The gene encoding the *Pho*Pol-II intein with Cys1Ala (C1A) mutation to inhibit self-cleavages during the purification was amplified from pSKDuet23 [19] using the following two oligonucleotides HK941: 5′-AGGATCCGGTAATGCCTTCCCGGGAGATACAAG and HK942: 5′-TGAAAGCTTACTGATGCGTCACAATATTTTC. The PCR product was cloned between *Bam*HI/*Hind*III of pHYRSF53 (Addgene #64696) to make a fusion protein with the H_6_-SUMO domain, resulting in plasmid pCARSF55D [19]. The plasmid pCARSF55D encodes the N-terminal H_6_-SUMO domain (yeast SMT3) fused with the *Pho*Pol-II intein with C1A mutation. The C-terminal residue was kept as the original glutamine (Gln), followed by the stop codon, resulting in no C-extein residue. The *Pho*Pol-II intein was produced in *Escherichia coli* strain T7 Express (New England Biolabs) using plasmids pCARSF55D and pRARE encoding for rare tRNAs to minimize the effect of codon bias in thermophilic organisms [18,20]. The transformed cells were grown at 37 °C in 2-L LB expression cultures supplemented with 25 μg mL^−1^ kanamycin and 5 μg mL^−1^ chloramphenicol. The cultures were induced with a final concentration of 1 mM isopropyl-β-D-thiogalactoside (IPTG) for 3 h when OD_600_ reached 0.6. The induced cells were harvested by centrifugation at 4700× *g* for 10 min, 4 °C and lyzed in 20-mL buffer A (50 mM sodium phosphate, pH 8.0, 300 mM NaCl) using an EmulsiFlex-C3 homogenizer (Avestin Inc, Ottawa, ON, Canada) at 15,000 psi for 10 min, 4 °C. Lysates were cleared by centrifugation at 38,000× *g* for 60 min, 4 °C. The *Pho*Pol-II intein (C1A) was purified using a 5-mL HisTrap HP column (GE Healthcare Life Sciences) as previously described, including the removal of the N-terminal H_6_-SUMO fusion domain [19]. The protein was dialyzed against deionized water and concentrated for crystallization using Macrosep^®^ Advance Centrifugal Devices 10K MWCO (PALL Corporation, New York, NY, USA).

### 2.2. Cis-Splicing of PhoPol-II Intein

For the *cis*-splicing test of *Pho*Pol-II intein, the gene of the active *Pho*Pol-II intein, including sequence encoding two residues of “GN” and “CD” at the N- and C-terminal splicing junction, was amplified from pCARSF55D as the template using the following two oligonucleotides, J603: 5′-AAGGATCCGGTAATTGCTTCCCGGGAGATACAA and J618: 5′-TAGGTACCATCGCACTGATGCGTCACAATATTTTC. The expression vector for the *cis*-splicing precursor with the two B1 domains of *Staphylococcus* protein A (GB1) was created by cloning the PCR product between *Bam*HI/*Kpn*I of SKDuet16, resulting in pLKRDuet30. The expression plasmid for the *cis*-splicing precursor protein with SUMO and chitin-binding domain (CBD) as N- and C-exteins was constructed by amplifying the intein gene using the following three oligonucleotides, M083: 5′-GAACAGATTGGTGGATCCAAACGTAATTGCTTCCCGGGAGATACAAGA, M084: 5′-GAACAGATTGGTGGATCCGCTAAGAAACGTAATTGCTTCCCGGGAGATACA, and M085: 5′-CACCAGGATTTGTGGTACCGTCGCACTGATGCGTCAC, followed by Gibson Assembly with pHYRSF53 using the following two additional oligonucleotides, J508: 5′-CGGTACCACAAATCCTGGTG and HB030: 5′-GGATCCACCAATCTGTTCTCTG, resulting in plasmid pJEJRSF294. The *cis*-splicing precursor protein with two GB1 domains as exteins or with SUMO and CBD were produced in *E. coli* strain T7 Express (New England Biolabs) using either plasmid pLKRDuet30 or pJEJDuet294 as described above. The precursor protein was purified using a 5-mL HisTrap HP column (GE Healthcare Life Sciences) or a Ni-NTA spin column and dialyzed against PBS. The purified precursor protein was incubated in the presence of 0.5 mM TCEP at either room temperature, 37, 50, or 60 °C. The samples were taken at 0, 1, and 3 h, and overnight (ON), and were analyzed by SDS-PAGE on 16.5% acrylamide gels and visualized using Coomassie blue staining. 

### 2.3. Crystallization of PhoPol-II Inteins

*Pho*Pol-II intein (C1A) (23.4 mg/mL) was used for crystallization trials. Drops of 200 nL (100 nL concentrated protein and 100 nL of reservoir solution) were set up in 96-well MRC (Molecular Dimensions) crystallization plates using a Mosquito LCP^®^ (TTP Labtech). Diffracting crystals were obtained with the reservoir solution containing 100 mM MES pH 6, 15% (*w/v*) PEG 550 MME, and 30 mM zinc sulfate. PEG MME 550 (25%) was added as a cryoprotectant for flash-freezing crystals in liquid nitrogen. The *Pho*Pol-II intein diffraction data were collected on beamline I03 at the diamond light source with Eiger2 XE 16M detector (Oxfordshire, UK) and were subsequently indexed, integrated, and scaled to a 1.48-Å resolution using the program XDS [21].

### 2.4. Structure Determination and Refinement 

The crystal structure of *Pho*Pol-II intein was solved by molecular replacement. The search model was modeled using SWISS-MODEL based on the primary structure [22]. The initial solution obtained from Phaser using the search model was used for auto-building by ARP/WARP [23]. The initial coordinates obtained from ARP/WARP were rebuilt with Coot, followed by rounds of refinement using the Phenix software [24,25]. The polypeptide chain of *Pho*Pol-II intein was fully traced into the electron density map without breaks for 168 residues, 166 of which belong to the intein. We used the comprehensive validation tool in the Phenix GUI for validating the quality of the final structure (Table 1) [25].

## 3. Results

### 3.1. Crystal Structure of PhoPol-II Intein

The DNA polymerase II large subunit (Uniprot: O57861) of *P. horikoshii* contains a 166 residue intein (*Pho*Pol-II intein). Previously, the Pol-II intein from *P. abyssi* (*Pab*Pol-II intein) has been solved by NMR spectroscopy [28]. Whereas *Pab*Pol-II consists of 185 residues, the *Pho*Pol-II intein only has 166 residues, which makes the latter more attractive for protein engineering due to its smaller size. The sequence identity between them is 64% (119/185 residues). We solved the three-dimensional structure of the *Pho*Pol-II intein at a 1.48-Å resolution by molecular replacement (Figure 1a,b). The crystal structure revealed the typical HINT fold with the β-strand insertion commonly observed among inteins from thermophilic organisms (Figure 1a) [29]. In line with the loop minimization often observed for proteins from thermophiles [30], we could trace the electron densities for all of the 168 residues without detecting any flexible linker sequences, which were present in the structure of the *Pab*Pol-II intein (Figure 1, Figure 2 and Figure 3) [28].

### 3.2. Comparison with Other Mini-Inteins from P. horikoshii

Although the three mini-inteins in *P. horikoshii* have similar sizes, between 166 and 172 residues, their three-dimensional structures show distinct differences (Figure 2). All three structures have the extended β-strand insertion (extβ) that is commonly observed among thermophilic inteins [29]. The helical lengths preceding extβ vary among the three structures, with 1.9, 3.6, and 4.2 helical turns for *Pho*Pol-II, *Pho*RadA, and *Pho*CDC21-1 inteins, respectively. The *Pho*RadA intein has a 10-residue insertion at the highly conserved HEN insertion site, which could be removed without affecting the splicing activity [17]. In contrast, the *Pho*Pol-II and *Pho*CDC21-1 inteins lack such insertions at the HEN insertion site (Figure 2). The relatively short helix of the *Pho*Pol-II intein is rather reminiscent of a canonical mesophilic intein. Despite their similar sizes, the structural comparison suggests that the three inteins are unlikely to have directly evolved from the same ancestral intein, due to the distributed short insertions and deletions over the entire sequence (Figure 2b).

### 3.3. Comparison with Naturally-Occurring Mini-Intein Structures from Other Thermophiles

We subjected the coordinates of the *Pho*Pol-II intein to the DALI protein structure comparison server in order to identify the closest three-dimensional structures. Not surprisingly, the server returned the *Pab*Pol-II intein (2LCJ) as the closest structure, with a Z-score of 22.7, covering 164 residues with 1.5 Å-RMSD for Cα atoms (Figure 3) [28]. The *Pab*Pol-II intein shares 70% sequence identity with the *Pho*Pol-II intein (Figure 1c). The largest difference between the two inteins lies in the flexible 19-residue sequence at the HEN insertion site for the *Pab*Pol-II intein (Figure 3), which is likely to be a remnant of HEN degradation during evolution. The structures of the CDC21-1 inteins from *P. horikoshii* and *P. abyssi* have Z-scores of 22.0 and 22.2, respectively [18]. These two inteins are closely related to the same insertion site in the CDC21 protein. However, the longer helices in the CDC21-1 inteins are very different from the Pol-II inteins. The observed variation in the helix length within the thermophilic insertion might be caused by differences in the evolutionary origins of the different inteins. The VMA mini-intein from *Thermoplasma volcanium* (*Tvo*VMA) has the same Z-score of 22.2. However, it resembles the *Pho*RadA intein because both structures share a prominent extension of the helix [14]. The KlbA intein from *Methanococcus jannaschii* (*Mja*KlbA) is another naturally-occurring mini-intein and has a Z-score of 19.1 with the *Pho*Pol-II intein [31]. All three-dimensional structures mentioned show the typical HINT fold, including the additional β-strand insertion (extβ) preceded by a helix of variable length, even though the growth temperatures of their hosts vary drastically between 33 and 104 °C (Table 2).

### 3.4. Protein Splicing Activity of PhoPol-II Intein

While the hyperthermophilic archaeon *Pyrococcus horikoshii* grows at a temperature between 88 and 104 °C (Table 2) [32], the *Pho*RadA intein is capable of efficiently catalyzing protein splicing at ambient (20–30 °C) temperatures in vitro and in *E. coli* cells [17]. This observation suggests that the protein splicing reaction is unlikely a rate limitation during the biosynthesis of the active RadA protein. At ambient temperatures (20–30 °C), the *Pho*CDC21-1 intein retains protein splicing activity with foreign exteins to a lesser extent than the *Pho*RadA intein in *E. coli*, but its splicing activity could be improved by increasing the temperature [19].

Conversely, the *Pho*Pol-II intein was found inactive without increasing the temperature [19]. At a higher temperature and longer incubation, the *Pho*Pol-II intein showed weak protein splicing activity as well as side reactions such as cleavages (Figure 4a–e). When we tested two different combinations of N- and C-exteins with the same junction sequences, the splicing activity and side-reaction profile were dissimilar (Figure 4d,e). The precursor proteins with a SUMO domain and chitin-binding domain (CBD) as exteins were less reactive than the precursor with two GB1 domains, even though the splicing junction sequences were identical. This observation indicates the extein dependency by inteins, presumably due to the mutualism between the intein and the host protein. 

Due to this delicate activity profile, it appears plausible that the *Pho*Pol-II intein might contribute to the physiological regulation of active DNA polymerase II production. In other words, the production of active DNA polymerase might be dependent on the protein splicing activity of the *Pho*Pol-II intein, which could be a survival strategy of mini-inteins lacking the HEN domain to protect themselves from elimination. 

Both the *Pho* and *Pab* Pol-II inteins have an atypical C-terminal Gln residue instead of the canonical Asn responsible for the cleavage of a branched intermediate during the protein splicing reaction (Figure 1c) [28,36]. The ^1^H-^15^N correlation peak for the C-terminal Gln was not visible for the *Pab*Pol-II intein in the HSQC spectrum, suggesting slow conformational exchanges rather than a fixed conformation [28]. The sidechain of the C-terminal Gln166 in the *Pho*Pol-II intein shows clear electron density, allowing the measurement of precise distances with other key residues (Figure 1b). His154 in block F presumably plays a crucial role in activating Gln cyclization for the cleavage of the branched intermediate. The distance between His154 and the carbonyl group of Q166 is 3.4 Å, which is shorter than the distance (4.6 Å) between the corresponding atoms of His160 in block F and the last residue, Asn172, in the *Pho*RadA intein structure (Figure 1b) [17]. It appears that His154 stabilizes the side-chain conformation of Q166, making it less reactive towards the side-chain cyclization of Q166. 

## 4. Discussion

Currently, there is a very limited number of inteins suitable for biotechnological applications because such inteins require (i) a most robust splicing activity in vivo and in vitro, (ii) fast reaction kinetics, (iii) a high tolerance of foreign extein contexts, and (iv) functional reconstruction of their catalytically active structures from split fragments in order to enable a versatile use of protein splicing [10,14,15]. Extremophiles could be good sources for hunting for new robust inteins because of the prevalence of inteins: nearly half of all archaea contain inteins [4,6,7]. Although more than 1,500 inteins have been identified, only a dozen mini-inteins have been biochemically characterized [7]. In the past, biochemical as well as structural studies of diversely or arbitrarily selected inteins were performed and resulted in serendipitously identifying robust inteins, which could be suitable for protein engineering [15,19]. We questioned whether there would be a good strategy for identifying mini-inteins with robust splicing activity from the fast-increasing genome sequence data prior to the extensive experimental assessment. 

Here, we determined the three-dimensional structure of a naturally-occurring mini-intein, *Pho*Pol-II intein, enabling us to compare the structures of all mini-inteins in the genome of *P. horikoshii*. Even though all three mini-inteins share the same HINT fold, with an extended β-strand insertion characteristic for inteins from thermophilic organisms [29], the three inteins show distinct differences in helical lengths and loop insertions. The distributed insertion and deletion differences for the entire sequences support the view of specific evolutionary pathways originating from unique ancestors. Among the three mini-inteins of *P. horikoshii,* only the *Pho*Pol-II intein requires an elevated temperature for some protein splicing activity in vitro, similarly to the *Pab*Pol-II intein [36]. The dependence of active DNA polymerase II production on protein splicing activity implies that the *Pho*Pol-II intein could play a critical role in regulating the fitness of the organism in response to environmental stimuli, such as temperature changes, because DNA polymerase II is an essential enzyme. With this strategy, mini-inteins like *Pho*Pol-II intein could avoid their removal under certain conditions [5]. The activity of the *Pho*Pol-II intein was also sensitive to the extein sequences in the same context of the N- and C-terminal junction sequences, suggesting that the mutualism with the host protein has been established so that the *Pho*Pol-II intein is more difficult to be eliminated during evolution. 

Mini-inteins lack HEN domains, either because they have been lost during evolution and/or because they have not yet been invaded by a homing endonuclease [13]. The closely related archaea *P. horikoshii* and *P. abyssi* contain 14 inteins in their genomes (Appendix A). Interestingly, the RadA intein with high splicing activity at room temperature is absent in *P. abyssi,* whereas both the CDC21 and Pol-II mini-inteins are present in both *P. horikoshii* and *P. abyssi*. The *Pho*RadA intein, which is highly capable of splicing at room temperature, might face a higher elimination pressure in hyperthermophilic organisms than the other mini-inteins requiring elevated temperatures for efficient splicing activity.

Most mini-inteins from extreme halophiles are not halo-tolerant but halo-obligatory inteins, meaning they require high salinity for protein splicing activity [12]. Mini-inteins must acquire a certain mutualism during evolution to become persistent, by providing certain post-translational benefits to the host to ensure survival under specific cellular and environmental conditions [5]. Indeed, salt-dependent inteins seem to give some advantages to the host organism [37]. Because of the established mutualisms with the host proteins, highly conserved inteins across a wide phylogenetic distribution might not be the most promising candidates for robust mini-inteins suitable for wider biotechnological applications. Such highly conserved mini-inteins residing at conserved insertion sites are likely to have developed a significant degree of mutualism in physiological conditions, eventually avoiding their removal under certain environmental or cellular conditions. Demonstrated examples include an elevated temperature for hyperthermophiles and a high salinity for extreme halophiles. Uncharacterized mini-inteins might provide some benefits to the host with other unknown regulatory functions, as found in the RadA intein [38]. Natural mini-inteins that are poorly conserved among different species and exist sporadically in the genome might be better candidates for further biochemical and structural characterizations. However, they might have already developed unknown reasons/regulatory functions to be persistent in their host organisms in physiological conditions, such as the mutualisms between mini-inteins and host proteins [39]. 

Alternatively, mini-inteins with efficient splicing activity could be obtained by artificially engineering mini-inteins by removing the HEN domains. More inteins bearing HEN domains than mini-inteins are found in genomic sequence databases, judging from their sizes [7,39]. Whereas some engineered inteins are fully capable of catalyzing protein splicing without the HEN domain, others have developed a mutualism with the HEN domain, requiring it for efficient protein splicing activity [9,14,15,16]. Unfortunately, it is unknown what makes their protein splicing activities dependent on the presence of a HEN domain. Thus, this approach still requires tedious experimental trials, including constructing several deletion variants [9,14,15,16]. Furthermore, only a few three-dimensional structures of inteins with HEN domains currently exist, imposing additional constraints on the rational design of HEN-free mini-inteins. Structural elucidation of inteins, particularly those with HEN domains, is awaited to better understand the structural basis of the mutualistic relationship between the HINT and HEN domains. Unveiling the structural basis of mutualism could help to expand the repertoire of promising mini-inteins by protein engineering in a rational and predictive way. 

In conclusion, we believe that it is vital to have a rational strategy for effectively selecting mini-intein candidates from the rapidly increasing genomic sequence data for experimental characterizations. Having a strategic approach to navigate the intein sequence space could efficiently increase the repertoires of mini-inteins with robust splicing activity, suitable for biotechnological applications. Our study suggests that sporadically distributed mini-inteins might be better candidates than conserved mini-inteins inserted in highly conserved proteins and locations. 

## Figures and Tables

**Figure 1 microorganisms-09-01226-f001:**
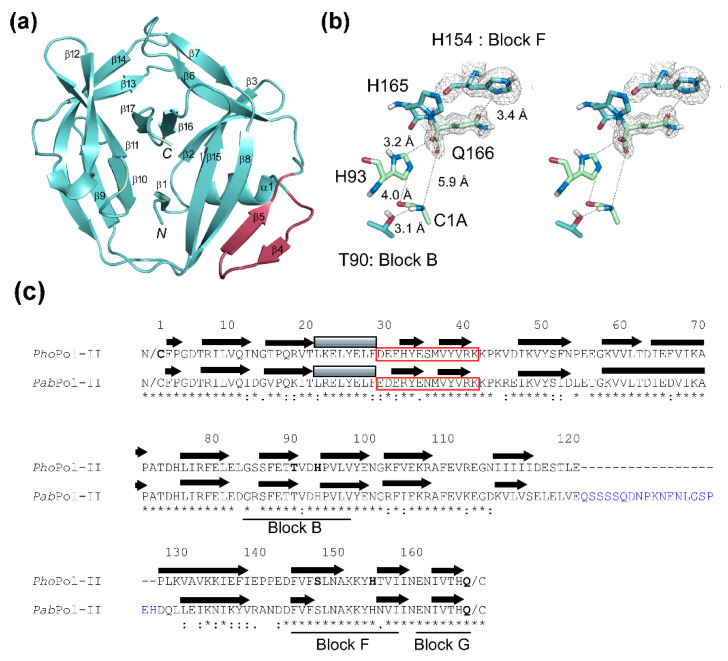
(**a**) A cartoon presentation of *Pho*Pol-II intein. The β-strand insertion common for thermophilic inteins is colored in orchard. *N* and *C* indicate the N- and C-termini, respectively. (**b**) A stereo view of the close-up of the active site showing Cys1Ala (C1A), Thr90 (T90), His93 (H93), His165(H165), and Gln166 (Q166). The electron densities are shown for H154 and Q166. (**c**) A sequence alignment of *Pho*Pol-II and *Pab*Pol-II inteins with secondary structure elements (arrows for b-strand and gray rectangles for helices). Red rectangles indicate the β-strand insertion. The flexible linker found in *Pab*Pol-II intein is colored in blue. Regions for blocks B, F, and G are marked by lines.

**Figure 2 microorganisms-09-01226-f002:**
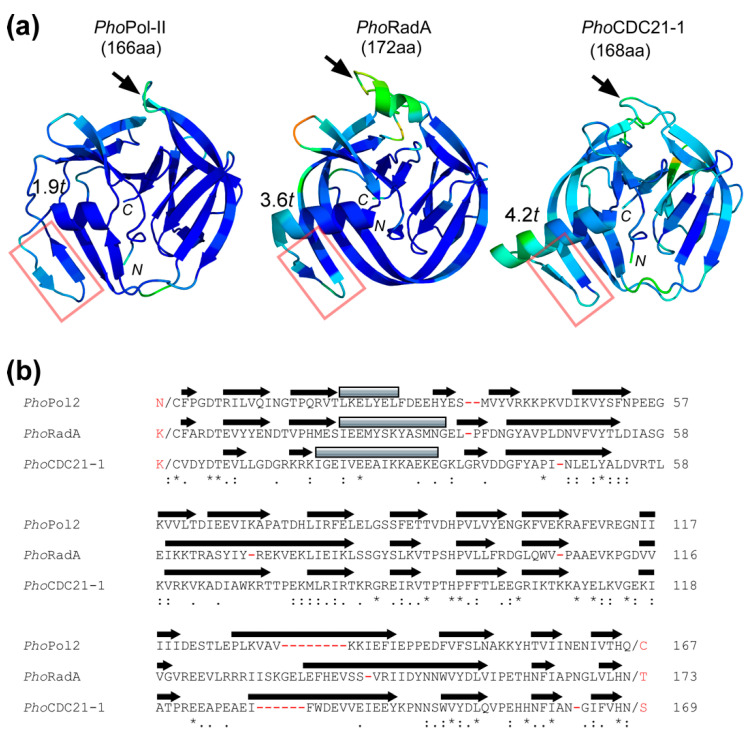
(**a**) Cartoon representations of the structures of *Pho*Pol-II (7OEC), *Pho*RadA (4E2T), and *Pho*CDC21-1 (6RPQ) inteins. The views are from the dorsal side [18]. Chain A from the coordinate (4E2T) was shown for *Pho*RadA intein [17]. *N* and *C* indicate N- and C-termini, respectively. The color codes represent the B-factor. The arrows indicate the insertion sites commonly observed for the homing endonuclease (HEN) domain. Red rectangles indicate the β-strand insertion common for thermophilic inteins. The length of helices is indicated by helical turns (*t*) (*t*: 3.6 residues per helical turn). (**b**) A sequence alignment of all three mini-inteins in *P. horikoshii* with secondary structure elements by arrows for β-strand and gray rectangles for helices. Deletions are shown by red hyphens. N- and C-extein residues are shown in red.

**Figure 3 microorganisms-09-01226-f003:**
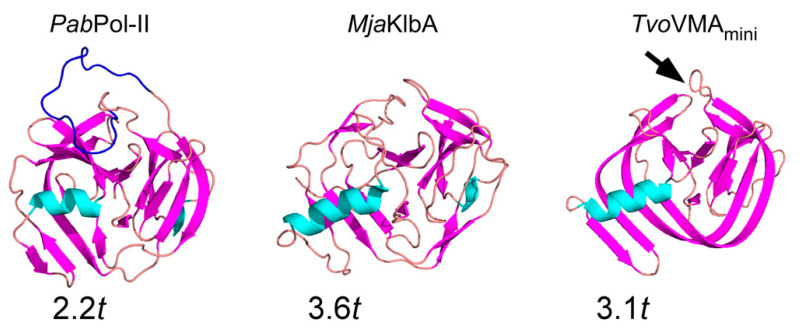
Cartoon models of the three-dimensional structures of the following mini-inteins from other thermophilic organisms: *Pab*Pol-II intein (**left**), *Mja*KlbA intein (**middle**), and *Tvo*VMA intein (**left**). PDB coordinates of 2LCJ (*Pab*Pol-II), 2JNQ (*Mja*KlbA), and 4O1S (*Tvo*VMA) were used. β-strands and helices are colored in magenta and cyan, respectively. The lengths of helices are indicated by the number of helical turns (*t*: 3.6 residues per helical turn). The flexible loop of *Pab*Pol-II is colored in blue. The location of the loop in *Tvo*VMA intein, of which 21 residues were removed for the crystallization, is indicated by an arrow.

**Figure 4 microorganisms-09-01226-f004:**
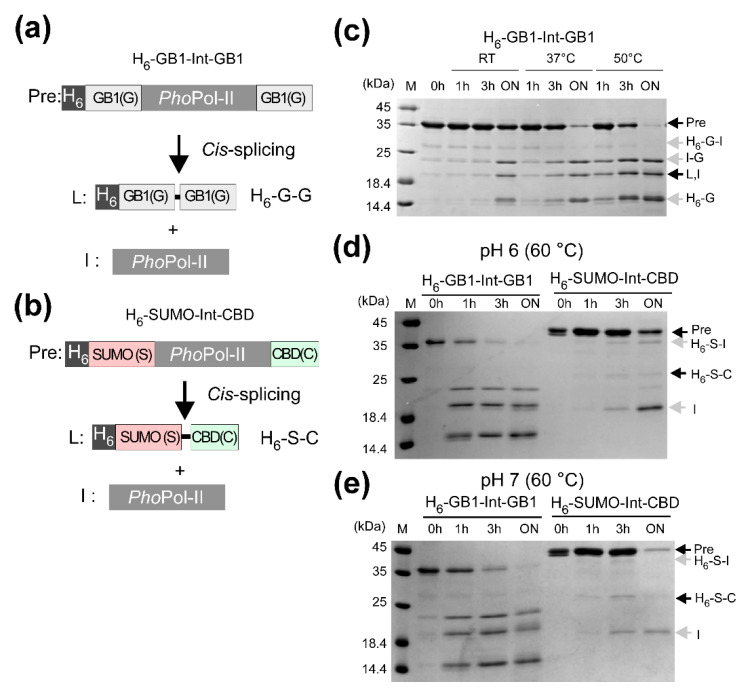
In vitro protein splicing in *cis* by *Pho*Pol-II intein with different exteins. (**a**) Precursor protein with two GB1 as exteins. (**b**) Precursor protein with SUMO domain and chitin-binding domain (CBD) as N- and C-exteins, respectively. Arrows indicate expected migrations of different products in SDS-PAGE gels. (**c**) The precursor protein (Pre) was purified and incubated at room temperature, 37 °C, and 50 °C. The two precursor proteins with different inteins were incubated at 60 °C and (**d**) pH 5 or (**e**) pH 7. The samples were taken at 0, 1, 3 h and overnight for SDS-PAGE analysis.

**Table 1 microorganisms-09-01226-t001:** Data collection and structure refinement.

Intein	*Pho*Pol-II Intein (C1A)
PDB ID **Data collection**	7OEC DIAMOND I03
Space group	P 4_1_ 2_1_ 2
Cell dimensions	
a, b, c, Å	70.82, 70.82, 70.66
α, β, γ, °	90.00, 90.00, 90.00
Wavelength, Å	0.9763
Resolution, Å	35.41−1.48 (1.53–1.48)
Total reflections	776,435 (76,171)
Unique reflections	30,544 (2978)
Completeness, %	99.87 (99.77)
I/σ	16.35 (4.23)
R_meas_ ^a^	0.1939 (7.025)
CC_1/2_^c^	0.998 (0.572)
Multiplicity	25.4 (25.5)
**Refinement**	
Molecules/*au*	1
Resolution, Å	35.41−1.48 (1.533–1.480)
Reflections (refinement/R_free_)	30,543/2978
R_work/_R_free_ ^b^	0.1539/0.1869
Number of atoms	
Protein	1382
Water	76
Ligand	34
RMS deviations	
Bond length, Å	0.017
Bond angles, °	1.44
Ramachandran plot, %	
Most favored regions	97.55
Outliers	0.00
Average B-factors, Å^2^	29.71
Protein	28.21
Water	37.29
Clash score	2.12
MolProbity score	0.97

Numbers in parentheses represent the highest resolution shell. *au*, asymmetric unit. ^a^
*R*_meas_ = Σ_h_[n/(n−1)]^1/2^ Σ_i_ ∣*I*_i_−⟨*I*⟩∣/Σ_h_Σ_i_
*I*_i_, where Ii is the observed intensity of the ith measurement of reflection h, ⟨I⟩ is the average intensity of that reflection obtained from multiple observations, and n is the multiplicity of the reflection. ^b^
*R* = Σ∣∣ *F**_o_*∣−∣*F**_c_*∣∣/Σ∣*F**_o_*∣, where *F*_o_ and *F*_c_ are the observed and calculated structure factors, respectively, calculated for all data. *R*_free_ was defined by Brünger [26]. ^c^ CC_1/2_ was defined by Karplus et al. [27].

**Table 2 microorganisms-09-01226-t002:** Summary of naturally-occurring mini-inteins from thermophiles and their growth temperatures.

Organism	Growth Temperature	Gene	PDB
*Pyrococcus horikoshii*	88–104 °C (98 °C) [32]	*radA*	2LQM, 4E2U, 4E2T
		*cdc21*	6RPQ
		*pol-c*	7OEC
*Pyrococcus abyssi*	68–102 °C (96 °C) [33]	*pol-c*	2LCJ
		*cdc21*	6RPP
*Thermoplasma volcanium*	33–67 °C (60 °C) [34]	*vma*	4O1S
*Methanococcus jannaschii*	48–94 °C (85 °C) [35]	*klbA*	2JMZ, 2JNQ

## Data Availability

The coordinates and structure factor of the *Pho*Pol-II intein have been deposited to the Protein Data Bank with accession number 7OEC.

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
