# Peer review of "Mini-Intein Structures from Extremophiles Suggest a Strategy for Finding Novel Robust Inteins"

_microorganisms, 2021, doi:10.3390/microorganisms9061226_

Round 1

Reviewer 1 Report

The paper reports a new mini-intein structure identified from Pyrococcus horikoshii and makes comparison with other inteins from other thermophiles. Few interesting structural features were raised among the comparison. Thus, to characterize inteins from different extremophiles might be a good strategy for identifying robust inteins for future particular usages.

Some suggestions should be considered to improve the paper.

Major:

Line 182: Despite the three inteins from Pho have similar size and slightly different structure fold (the helix), the author make the conclusion that the inteins unlikely to be evolved from the same ancestral intein. However, the explanation is relatively weak at least from the attached information. One helix with variable lengths might not be enough to derive the conclusion. At least sequence comparison should be included here.

Figure 4: The SDS-PAGE is the only data to explain the intein activity. The bands corresponding to L, I seem to be merged here. However, we cannot distinguish whether the ligated product (L) is indeed produced, or Pol-II intein only can perform N- and C-terminal cleavages to cause fraction, I. It might need better explanation or other data to support it.

Line 223-238: The description from the lines comes out no focusing. The dynamics of the C-terminal Gln of Pab and Pho Pol-II inteins might be different. But then? No explanation or further speculation make this part unfinished.

In summary, the paper brought a high-quality structure of intein. However, only one structure makes less contribution for the field. The author should consider to have a deep comparison to different inteins of extremophiles, based on their structures. Or, the paper can put more emphasis on clarifying the activity of Pol-II under different conditions (temp, pH, salt). The lab already made many extraordinary contributions in intein field. The readers might expect to learn more from this paper.

Minor:

Line 146: It is wired to suddenly jump to Figure 3 from Figure 1. Consider to modify the writing.

Figure 2: Please add PDB codes for the structures.

Line 143, 174, 204: Where is the extended b-strand insertion? Possible to indicate on the structures?

Line 192: It should be PabPol-II?

Table 2: The gene pol-c is pol-II?

Author Response

Major: Line 182: Despite the three inteins from Pho have similar size and slightly different structure fold (the helix), the author make the conclusion that the inteins unlikely to be evolved from the same ancestral intein. However, the explanation is relatively weak at least from the attached information. One helix with variable lengths might not be enough to derive the conclusion. At least sequence comparison should be included here.

Response: We included the sequence alignment as Figure 2b. Figure 2b highlights various short deletions scattered along the sequences to support our claim. It is also noteworthy that sequence conservation is also not very high among the three mini-inteins in P. horikoshii.

  • Figure 4: The SDS-PAGE is the only data to explain the intein activity. The bands corresponding to L, I seem to be merged here. However, we cannot distinguish whether the ligated product (L) is indeed produced, or Pol-II intein only can perform N- and C-terminal cleavages to cause fraction, I. It might need better explanation or other data to support it.

Response: Indeed, it is unclear about the ligation and/or cleavages because precipitations of different fragments took place during the heat incubations and similar migration in SDS-PAGEs. Thus, we did not discuss the detail such as kinetics or efficiencies. However, we also included another analysis using precursors with different exteins in the identical junction-residue context (two GB1 vs SUMO and CBD). The reaction seems to be worse for the precursor with SUMO and CBD exteins despite the identical junction sequences. This observation also supports our claim that PhoPol-II intein has already developed mutualism with the host protein and not a suitable candidate for protein engineering.

  • Line 223-238: The description from the lines comes out no focusing. The dynamics of the C-terminal Gln of Pab and Pho Pol-II inteins might be different. But then? No explanation or further speculation make this part unfinished.

Response: This is a good question if there are some differences with protein dynamics at this residue between Pab and PhoPol-II.  We substantially revised this part.

  • In summary, the paper brought a high-quality structure of intein. However, only one structure makes less contribution for the field. The author should consider to have a deep comparison to different inteins of extremophiles, based on their structures. Or, the paper can put more emphasis on clarifying the activity of Pol-II under different conditions (temp, pH, salt). The lab already made many extraordinary contributions in intein field. The readers might expect to learn more from this paper.

Response: We appreciate the comments. Extensive mutational and conditional studies on the homologous PabPol-II to delineate the mechanism, which has about 70% identity with PhoPol-II intein, have already been reported, for example, by Mills et al. JBC, 2004 [ref. 34]. Nevertheless, our understanding of the splicing mechanism is still limited.

 We believe that the three-dimensional structure presented here could provide vast information as the coordinate towards understanding the structure-function relationship. This work completes the set of three mini-inteins structures in P.horikoshii, out of all 14 identified inteins in the genome of P.horikoshii. The structural comparison within P.horikoshii and with other natural mini-inteins has led to a possible strategy to select good candidates from the fast-increasing intein sequence space.

Robust mini-inteins with fast kinetics in foreign contexts are highly desirable for various applications in synthetic biology and biotechnology. However, they are very limited despite >1500 inteins identified in the genome sequence data. Here we propose a strategy to select inteins for experimental characterizations from our studies, instead of focusing on the mechanistic aspect of one intein, which was already reported by others in the detail. Thus, the mechanistic detail is not the scope of this article.  We emphasized our claim in the discussion.

Minor:

  • Line 146: It is wired to suddenly jump to Figure 3 from Figure 1. Consider to modify the writing.

 Response: We changed the figures with some additions as requested by the reviewers. Now they do not jump from Figure 1 to 3.

  • Figure 2: Please add PDB codes for the structures.

Response: We added.

  • Line 143, 174, 204: Where is the extended b-strand insertion? Possible to indicate on the structures?

 Response: We indicated the extended beta-strand insertion by red rectangles in Figures 1 and 2.

  • Line 192: It should be PabPol-II?

 Response: We fixed the error.

  • Table 2: The gene pol-c is pol-II?

Response: According to the UniProt database (O57861). The gene name for this host protein is given “pol-c”.  We leave the annotation in the UniProt database.

Reviewer 2 Report

Dear Editor,

In the manuscript entitled “Mini-intein structures from extremophiles suggest a strategy for finding novel robust inteins”, Hiltunen and co-authors solved the 3D structure of a mini-intein identified in the PolII gene from Pyrococcus horikoshii. The paper is well written, and the conclusions are supported by the results, however an overview of the 14 inteins of Pyrococcus horikoshii might be useful for not-expert readers.

Pag. 1 Line 15. Pyrococcus horikoshii should be P. horikoshii

Pag. 1 Line 31. The role and abundance of inteins in extremophiles should be described in more detail. It is not clear whether the inteins are present only in thermophiles and halophiles or are also present in other extremophiles such as psychrophiles, acidophiles and alkalophiles.

Pag. 2 Line 67. A figure containing the architecture of the 14 inteins of P. horikoshii could be useful to better appreciate the similarity and differences of these inteins.

Pag. 2 Line 78. Please highlight the BamHI and KpnI restriction sites in oligonucleotides.

Pag. 2 Line 79. Which plasmid did the authors use for cloning? Please indicate the differences between pHYRSF53 and pCARSF55D.

Pag.2 Line 84. E. coli should be Escherichia coli.

Pag. 2 Line 85. Please briefly described the plasmid pRARE.

Pag 3. Line 101. It is unclear why the Authors named the oligonucleotides with: J603, J618, HK941 and HK942. Please clarify.

Pag. 3 Line 105. See the comment Pag.2 Line 79.

Pag. 3 Line 137. It is unclear how the authors found the intein in the large subunit of DNA polymerase II. Is this intein present only in DNA polymerase II or is it present in other proteins? Please clarify.

Pag. 3 Line 139. Please add the sequence identity between PabPol-II and PhoPol-II.

Pag. 3 Line 145. What do the authors mean by “apparent thermophilic structure minimization”? Please clarify.

Figure 2. Please add in the caption the PDB code of PhoCDC21-1 and the meaning of 1.7t, 3.6t and 4.2t.

Pag. 6 Line 221. Please indicate the temperature in °C of the ambient temperature.

Figure 3a. The secondary structure of PhoPolII extracted from the PDB file should be shown above the alignment.

Author Response

In the manuscript entitled “Mini-intein structures from extremophiles suggest a strategy for finding novel robust inteins”, Hiltunen and co-authors solved the 3D structure of a mini-intein identified in the PolII gene from Pyrococcus horikoshii. The paper is well written, and the conclusions are supported by the results, however an overview of the 14 inteins of Pyrococcus horikoshii might be useful for not-expert readers.

Response: Inteins have increasingly become useful tools in protein engineering, synthetic biology, and biotechnology. However, mini-inteins with highly robust splicing activity are very limited, restricting the wider applications of inteins.  Almost half of the archaea including extremophiles contain up to 16 host proteins with inteins, although most of the organisms contain only a few proteins bearing inteins (Novikova et al. MBE 2016, Geen et al. Mobile DNA 2018).  In the past, highly conserved inteins and arbitrarily selected inteins were subjected to biochemical and structural characterization.  Thus, the identification of robust inteins has been relying on serendipitous discoveries of robust inteins. We now completed structural elucidation of all three mini-inteins in P. horikoshii, allowing us to compare within P.horikoshii as well as other thermophiles. Our conclusion is that on contrary to the current practice, non-conserved and sporadically found mini-inteins could be better candidates for biochemical characterization to identity robust inteins. This strategy could accelerate the discoveries of robust inteins suitable for biotechnological applications without relying on serendipitous discoveries.  We believe that our article fits well with the scope of this special issue, "Microbial Extremophiles as Life Pioneers and Wellsprings of Valuable Molecules", which serves to highlight the microbial biodiversity in extreme habitats and the unusual properties of extremophile molecules for exploitation by biotech companies.

We emphasized our claim at the end of the discussion.

  • Pag. 1 Line 15. Pyrococcus horikoshii should be P. horikoshii

Response: We replaced.

  • Pag. 1 Line 31. The role and abundance of inteins in extremophiles should be described in more detail. It is not clear whether the inteins are present only in thermophiles and halophiles or are also present in other extremophiles such as psychrophiles, acidophiles and alkalophiles.

Response: The distributions of inteins have been extensively investigated by several publications, such as Novikova et al. MBE (refs. 4,5,6,7,40). The distribution in different extremophiles is outside of the scope of this article, which aims to propose a strategy to find a highly robust mini-inteins, navigating the rapidly increasing sequence space of inteins in the genomic sequence data. We extended our introduction referring their comparative studies of genome sequences as well as Table S1. We also emphasized our claim in the conclusion.

  • Pag. 2 Line 67. A figure containing the architecture of the 14 inteins of P. horikoshii could be useful to better appreciate the similarity and differences of these inteins.

Response: Only three structures of inteins from P. horikoshii are available and presented in this article.  Three-dimensional structures of other inteins having the HEN domains are not available yet.  We included Table S1 summarizing the sizes of inteins in a few thermophilic organisms.

  • Pag. 2 Line 78. Please highlight the BamHI and KpnI restriction sites in oligonucleotides.

Response: We underlined the sequence and fixed an error.

  • Pag. 2 Line 79. Which plasmid did the authors use for cloning? Please indicate the differences between pHYRSF53 and pCARSF55D.

Response: pHYRSF53 is the backbone plasmid to create a SUMO fusion protein with the intein. The resulted plasmid bearing SUMO-intein fusion is pCARSF55D.  We rephrased.

  • Pag.2 Line 84. E. coli should be Escherichia coli.

Response: We changed.

  • Pag. 2 Line 85. Please briefly described the plasmid pRARE.

Response: We described.

  • Pag 3. Line 101. It is unclear why the Authors named the oligonucleotides with: J603, J618, HK941 and HK942. Please clarify.

Response: These names are our arbitral names used in the group and also used in our all publications. This allows reproducing our work for those who follow our publications. The clarification of the names has no practical use for general readers.

  • Pag. 3 Line 105. See the comment Pag.2 Line 79.

Response: We changed.

  • Pag. 3 Line 137. It is unclear how the authors found the intein in the large subunit of DNA polymerase II. Is this intein present only in DNA polymerase II or is it present in other proteins? Please clarify.

Response: The identification of inteins in DNA polymerases have been repeatedly reported in the past . It was also extensively analyzed and reported by other such as Novikova et al. MBE 2016 (refs, 4-7 and others).  We added Table S1 to show some distributions from Pietrokovski et al. 1997 (ref. 6). 

  • Pag. 3 Line 139. Please add the sequence identity between PabPol-II and PhoPol-II.

Response: We added the sequence identity.

  • Pag. 3 Line 145. What do the authors mean by “apparent thermophilic structure minimization”? Please clarify.

Response: we rephrased the sentence.

  • Figure 2. Please add in the caption the PDB code of PhoCDC21-1 and the meaning of 1.7t, 3.6t, and 4.2t.

Response: We added the definition of “t” in the caption.

  • Pag. 6 Line 221. Please indicate the temperature in °C of the ambient temperature.

Response: We indicated. 

  • Figure 3a. The secondary structure of PhoPolII extracted from the PDB file should be shown above the alignment.

Response: We added this as in Figure 1c.

Round 2

Reviewer 1 Report

The current manuscript is with good improvement and good for accepted.